# Face masks while exercising trial (MERIT): a cross-over randomised controlled study

Nicholas Jones ,[1] Jason Oke,[1] Seren Marsh,[2] Kurosh Nikbin,[3] Jonathan Bowley,[4] H Paul Dijkstra,[5,6] FD Richard Hobbs,[1] Trisha Greenhalgh [1]

[1]Nuffield Department of Primary Care Health Sciences, University of Oxford, Oxford, UK
[2]University of Oxford Medical School, University of Oxford, Oxford, UK
[3]GKT School of Medical Education, King's College London, London, UK
[4]School of Medicine, University of Nottingham, Nottingham, UK
[5]Department of Continuing Education, University of Oxford, Oxford, UK
[6]Medical Education Department, Aspetar Orthopaedic and Sports Medicine Hospital, Doha, Ad Dawhah, Qatar

**Correspondence to**
Dr Nicholas Jones;
nicholas.jones2@phc.ox.ac.uk

## ABSTRACT

**Objectives** Physical exertion is a high-risk activity for aerosol emission of respiratory pathogens. We aimed to determine the safety and tolerability of healthy young adults wearing different types of face mask during moderate-to-high intensity exercise.

**Design** Cross-over randomised controlled study, completed between June 2021 and January 2022.

**Participants** Volunteers aged 18–35 years, who exercised regularly and had no significant pre-existing health conditions.

**Interventions** Comparison of wearing a surgical, cloth and filtering face piece (FFP3) mask to no mask during 4×15 min bouts of exercise. Exercise was running outdoors or indoor rowing at moderate-to-high intensity, with consistency of distance travelled between bouts confirmed using a smartphone application (Strava). Each participant completed each bout in random order.

**Outcomes** The primary outcome was change in oxygen saturations. Secondary outcomes were change in heart rate, perceived impact of face mask wearing during exercise and willingness to wear a face mask for future exercise.

**Results** All 72 volunteers (mean age 23.9) completed the study. Changes in oxygen saturations did not exceed the prespecified non-inferiority margin (2% difference) with any mask type compared with no mask. At the end of exercise, the estimated average difference in oxygen saturations for cloth mask was −0.07% (95% CI −0.39% to 0.25%), for surgical 0.28% (−0.04% to 0.60%) and for FFP3 −0.21% (−0.53% to 0.11%). The corresponding estimated average difference in heart rate for cloth mask was −1.20 bpm (95% CI −4.56 to 2.15), for surgical 0.36 bpm (95% CI −3.01 to 3.73) and for FFP3 0.52 bpm (95% CI −2.85 to 3.89). Wearing a face mask caused additional symptoms such as breathlessness (n=13, 18%) and dizziness (n=7, 10%). 33 participants broadly supported face mask wearing during exercise, particularly indoors, but 22 were opposed.

**Conclusion** This study adds to previous findings (mostly from non-randomised studies) that exercising at moderate-to-high intensity wearing a face mask appears to be safe in healthy, young adults.

**Trial registration number** NCT04932226

## INTRODUCTION

SARS-CoV-2 is an airborne virus that causes COVID-19. Transmission of SARS-CoV-2 is thought to vary based on multiple factors, including host viral load, indoor versus

### STRENGTHS AND LIMITATIONS OF THIS STUDY

⇒ This is the largest study to date assessing the safety and tolerability of exercising wearing a face mask.
⇒ The randomised cross-over design ensured that the effect of any fatigue from exercise was equal between mask types.
⇒ Some participants exercised outdoors and during cold weather when pulse oximeters were sometimes slow to provide readings, and this may have resulted in a recovery in oxygen levels and an underestimate of effect size.
⇒ We collected a limited amount of non-invasive physiological data and did not collect data on measures such as end-tidal carbon dioxide or muscle oxygenation.
⇒ We only included healthy young adults who exercised regularly in this study to minimise the risk of serious adverse events, but this may limit the generalisability of our findings to the wider population.

outdoor settings, ambient air flow, volume of speech and physical proximity.[1] A number of these risk factors are relevant to exercise, with previous outbreak clusters reported in boxing or fitness gyms, where people are likely to be talking loudly or breathing heavily in a confined, indoor space.[2 3] Public health measures such as physical distancing can help reduce transmission risk,[4] but heavy breathing or panting during exercise may still increase the risk of transmitting the SARS-CoV-2 virus, even when people are exercising outdoors.[1]

Face masks can reduce transmission of SARS-CoV-2 (box 1). In a recent systematic review and meta-analysis, mask wearing was associated with a decrease in the incidence of COVID-19 (relative risk 0.47, 95% CI 0.29 to 0.75, $I^2$=84%).[5] However, there has been a lack of clear guidance on wearing face masks during exercise and a number of recent commentary pieces have suggested that face masks may interfere with an individual's exercise capacity.[6] Exercise leads to physiological changes, such as an increase in heart and

### Box 1   Properties of different kinds of face covering

A filtering face piece respirator (FFP) is personal protective equipment made to an industry standard and designed for occupational health use (eg, healthcare workers treating high-risk infectious patients). FFP3 (equivalent to the US standard N99) filters 99% of airborne particles of the relevant size.

A surgical mask is made of folded paper with waterproof backing. It is designed primarily to block egress (outward emission) of droplets such as saliva or sneezes. Surgical masks are not made to industry standards, nor are they designed to filter airborne particles. Poor fit is a commonly described problem with surgical masks, allowing air to escape through gaps.

The cloth face covering used in this study was a double-layer neck gaiter supplied by Buff. Cloth face coverings come in many materials and thicknesses; most have not been manufactured to any filtration standard.[23] They are thought to provide partial protection of others by reducing egress of particles and droplets.[24]

The efficacy of masks depends heavily on fit and compliance.[16 25] Despite their limited efficacy, both surgical masks and cloth face coverings may contribute significantly to reducing SARS-CoV-2 transmission if a high proportion of the population wears them in high-risk settings (eg, indoors, crowded and at times of high prevalence of COVID-19).[26 27] In one laboratory study, a respirator reduced filtration of an artificial respiratory aerosol by 99%; a medical grade ('surgical') mask by 59% and a double neck gaiter by 60%.[28] A modelling study calculated that if two people are both wearing respirators, the risk of infection is reduced to 0.14% of the risk if neither are masked.[16]

breathing rates, to deliver more oxygen to the body. A face mask may theoretically interfere with gas exchange, potentially leading to an accumulation of carbon dioxide ($CO_2$) and lactate, with a reduction in oxygen saturation ($SaO_2$). This may depend in part on the type of mask worn.[7 8] Cloth masks tend to capture moisture in exhaled breath to a greater extent than surgical or filtering face piece (FFP) respirator masks, which may particularly interfere with oxygen transfer.[9]

There have been a number of recent studies assessing the impact of wearing an FFP or surgical mask during exercise, but most of these included very small numbers of participants.[10 11] As health and fitness facilities and physical therapy rehabilitation units reopen, it is important to determine to what extent wearing a face mask in these settings is feasible, in addition to other public health measures. We aimed to determine the safety and tolerability of healthy young adults wearing different types of face mask during moderate-to-high intensity exercise.

## METHODS
### Study design
We conducted a randomised cross-over study, where all participants completed 4×15 min bouts of moderate-to-high intensity exercise wearing three different types of face mask (cloth, surgical or FFP3) and no face mask in a random order, with a 5 min rest period between each bout of exercise. The study team used an online simple random sequence generator to allocate mask order for

each participant, without blinding. Participants could state a preference for their choice of exercise, between either running, cycling or rowing.

### Participants
People in the community were eligible for the study if they were aged between 18 and 35 years, exercised at least three times a week and were willing and able to give informed consent. We excluded anyone with a significant acute or longstanding medical illness that limited their exercise capacity. Volunteer participants were recruited through student groups at the University of Oxford, Nottingham University and King's College London.

### Sample size
When planning the study, there was only limited existing research of changes in $SaO_2$ during exercise. A 2018 study assessed the effect of aerobic exercise on arterial blood $SaO_2$ among 36 healthy male volunteers. Mean pre-exercise $SaO_2$ was 97.3% (SD 1.3, range 95.0%–99.0%), compared with mean postexercise $SaO_2$ 96.2% (2.2, 92.0%–98.0%).[12] We made a conservative estimate that exercise with a face mask might produce a 1.5% margin of difference in peripheral $SaO_2$. Based on an SD of 2.6 and power of 80%, we calculated a minimum sample size of 55 people was needed, but aimed to recruit over 60 to allow for some participants dropping out and a 10% margin of error.

### Outcomes
The primary outcome was change in $SaO_2$ between different mask types. Secondary outcomes were change in heart rate during exercise, perceived impact of mask wearing during exercise and willingness to wear a mask for exercise in the future.

### Data collection
$SaO_2$ and heart rate were measured using a finger pulse oximeter immediately prior to first commencing exercise, halfway through, on completion and 1 min after completion for each bout of exercise. Participants paused their exercise at the midway point of exercise to collect these data. Participants continued to wear their face mask for 1 min after completing the exercise, when the final recording for that bout of exercise was collected. Participants were asked to exercise at a consistent intensity between bouts of exercise. To confirm this was achieved, we measured the average speed and distance travelled during each bout of exercise using a Smartphone app (Strava) and assessed these data for significant differences across face mask type.

During each 5 min rest period, the participants stood and completed a study questionnaire. This asked them to rate the comfort, ease of exercise and ease of breathing during exercise with the face mask they had just worn on a 10-point Likert scale. At the end of the study session, we also asked participants to identify the face mask that they found most difficult to exercise wearing, record any symptoms that they felt during exercise that were different to

usual and free text comments regarding their willingness to exercise wearing a mask in the future (see online supplemental appendix 1).

## Statistical analysis

We calculated the difference in $SaO_2$ and heart rate (bpm) between exercising with each of the three different face masks compared with no mask in each participant. We used differences rather than raw outcomes as these are more likely to be normally distributed. Dependency within subjects was accounted for by including a subject specific effect in a two-way analysis of variance. Average differences and 95% CIs for the three comparisons (cloth vs none, surgical mask vs none and FFP3 vs none) were calculated using estimated marginal means, averaging over subject. Based on prior literature, we prespecified a non-inferiority margin of a 2% reduction in $SaO_2$ and 7 bpm increase in heart rate. Non-inferiority was considered demonstrated if the lower bound of the 95% CI did not overlap the 2% margin for $SaO_2$ and if the upper bound of the 95% CI did not overlap 7 bpm for heart rate. We conducted a subgroup analysis to check for significant changes in $SaO_2$ or heart rate among participants with a low baseline $SaO_2$, defined as ≤98%. Comparisons of average distance travelled and ease of breathing while wearing different types of face mask compared with no mask were analysed using a Bradley-Terry Model for Paired Preferences.[13] All analyses were done in R V.4.1.1 (2021-08-10).

## Qualitative analysis

Free-text comments were grouped into categories using a predefined framework, based around positive, negative or neutral comments for each type of mask. Using a comment-by-comment coding approach in Excel, we then undertook thematic analysis to identify and group common threads within these categories, seeking emergent as well as anticipated themes.[14 15]

## Patient and public involvement

No patients involved.

## RESULTS

In total, 72 participants completed the study between June 2021 and January 2022, including 33 women (46%) and 39 men (54%). The mean age was 23.9 years (SD 3.55). Most participants exercised 3–4 (n=38, 54%) or 5–6 times per week (n=30, 42%), but 3 participants (4%) exercised every day. One participant was a smoker.

All 72 participants completed all four exercise sessions. One participant completed the exercise on an indoor rowing machine and 71 undertook running outdoors, largely due to restrictions on indoor gatherings and physical distancing that were in place because of the COVID-19 pandemic.

In a small number of participants at each time point, it was not possible to obtain either a $SaO_2$ or heart rate reading on the pulse oximeter and the numbers reported in tables 1 and 2 and the supplemental tables reflect this. We recruited beyond our minimum sample size to ensure the study retained sufficient statistical power for our primary analysis.

## $SaO_2$ during exercise

Changes in $SaO_2$ did not exceed the non-inferiority margin at any point of exercise wearing any of the three masks (table 1). There was a small but statistically significant decrease in $SaO_2$ during exercise wearing either a cloth mask (estimated average difference (EAD) −0.78%, 95% CI −1.12% to −0.44%) or the FFP3 compared with no mask (−0.82%, 95% CI −1.16% to −0.47%), but not

**Table 1** Changes in oxygen saturations ($SaO_2$) comparing different types of face mask worn during exercise

| Comparison | Average oxygen saturation (%), with mask | Average oxygen saturation (%), no mask | Estimated average difference (%, 95% CI) | Non-inferior at 2% difference in $SaO_2$ |
|---|---|---|---|---|
| **During exercise (N=69)** | | | | |
| Cloth—none | 96.9 | 97.7 | −0.78 (−1.12 to −0.44) | True |
| Surgical— none | 97.4 | 97.7 | −0.28 (−0.63 to 0.07) | True |
| FFP3—none | 96.9 | 97.7 | −0.82 (−1.16 to -0.47) | True |
| **End of exercise (N=68)** | | | | |
| Cloth—none | 97.0 | 97.1 | −0.07 (−0.39 to 0.25) | True |
| Surgical—none | 97.4 | 97.1 | 0.28 (−0.04 to 0.60) | True |
| FFP3—none | 96.9 | 97.1 | −0.21 (−0.53 to 0.11) | True |
| **1 min after the end of exercise (N=67)** | | | | |
| Cloth—none | 97.7 | 97.7 | 0.04 (−0.25 to 0.32) | True |
| Surgical—none | 98.1 | 97.7 | 0.33 (0.05 to 0.61) | True |
| FFP3—none | 97.9 | 97.7 | 0.15 (−0.13 to 0.43) | True |

FFP, filtering face piece.

**Table 2** Changes in heart rate comparing different types of face mask worn during exercise

| Comparison | Average heart rate (bpm), with mask | Average heart rate (bpm), no mask | Estimated average difference (bpm, 95% CI) | Non-inferior @ +7 bpm |
|---|---|---|---|---|
| **During exercise (N=69)** | | | | |
| Cloth—none | 157.5 | 157.3 | 0.10 (−2.90 to 3.10) | True |
| Surgical—none | 157.8 | 157.3 | 0.78 (−2.33 to 3.89) | True |
| FFP3—none | 159.9 | 157.3 | 2.84 (−0.24 to 5.91) | True |
| **End of exercise (N=69)** | | | | |
| Cloth—none | 156.9 | 158.2 | −1.20 (−4.56 to 2.15) | True |
| Surgical—none | 158.7 | 158.2 | 0.36 (−3.01 to 3.73) | True |
| FFP3—none | 158.9 | 158.2 | 0.52 (−2.85 to 3.89) | True |
| **1 min after the end of exercise (N=67)** | | | | |
| Cloth—none | 124.8 | 124.2 | 0.40 (−2.15 to 2.95) | True |
| Surgical—none | 124.6 | 124.2 | 0.49 (−2.03 to 3.02) | True |
| FFP3—none | 125.3 | 124.2 | 1.45 (−1.11 to 4.01) | True |

FFP, filtering face piece.

when wearing the surgical mask (−0.28%, 95% CI −0.63% to 0.07%). There was no difference in $SaO_2$ comparing any mask type to no mask immediately after completing exercise. Surgical mask wearing was associated with a small increase in $SaO_2$ at 1 min post exercise (0.33%, 95% CI 0.05% to 0.61%).

In a subgroup analysis among those with lower baseline $SaO_2$, changes in $SaO_2$ still did not exceed the non-inferiority margin of 2% although $SaO_2$ during exercise was 1% lower among people wearing either the FFP3 (EAD −0.88, 95% CI −1.47 to −0.29) or cloth mask (−0.90, 95% CI −1.48 to −0.32) (online supplemental table 1a). Even those with $SaO_2 > 98\%$ at baseline had a small but statistically significant decrease in $SaO_2$ during exercise when wearing either the cloth or FFP3 mask (online supplemental table 1b).

### Changes in heart rate, performance or additional symptoms during exercise

There was no difference in heart rate at any time point comparing the three types of mask to no mask (table 2) and changes in heart rate did not exceed the non-inferiority margin.

Among the subgroups with $SaO_2 \leq 98\%$ or $SaO_2 > 98\%$ at baseline, there was no significant difference in average heart rate during or after exercise with any face mask (online supplemental tables 2a and b). However, for the subgroup with $SaO_2 \leq 98\%$ the upper 95% CI did exceed the non-inferiority margin of 7 bpm for the FFP3 mask compared with no mask at each time point measured, although the sample size for this analysis was small. This was also true for the surgical mask at the end of exercise and after a 1 min rest period (online supplemental table 2a). For the subgroup with $SaO_2 > 98\%$ the upper 95% CI exceeded the non-inferiority margin for the FFP3 and surgical masks compared with no mask during exercise only (online supplemental table 2b).

A post hoc sensitivity analysis excluding the one person who rowed rather than ran did not significantly alter the results for change in $SaO_2$ or heart rate (online supplemental table 3a and b).

Across participants, there was no significant difference in average speed or distance travelled when comparing any of the face masks to no mask, confirming participants had been able to exercise at a fairly consistent speed between bouts of exercise (table 3).

The most common symptoms participants reported feeling wearing a face mask during exercise were increased breathlessness (n=13, 18%), dizziness (n=7, 10%), 'other' symptoms (n=7, 10%) and fatigue (n=6, 8%). The 'other' symptoms recorded as free-text comments were feeling light-headed, facial discomfort, skin irritation, a runny nose, pressure in the eyes, cramping and a stitch. Four participants found wearing a mask claustrophobic and one developed a headache. No participants reported feeling anxious or drowsy and none had to stop during exercise due to side effects.

### Perception of comfort, breathing and ease of exercise

When asked which face mask they found most difficult to exercise wearing, 40 (56%) participants said the cloth mask, 27 (38%) said FFP3 and 4 (6%) said the surgical mask. Participants felt that the surgical mask was the most comfortable (figure 1) and the easiest to breathe through (figure 2). In a direct comparison, the probability of a surgical mask being ranked easier to exercise in than a cloth mask was 0.81 (95% CI 0.72 to 0.88, p=0.00) and being easier than FFP3 was 0.80 (95% CI 0.73 to 0.86, p=0.00) (table 4). Surgical masks were also ranked easier to breathe through than the cloth mask (probability 0.86, 95% CI 0.78 to 0.91) or FFP3 (0.83, 95% CI 0.76 to 0.88). The FFP3 and

**Table 3** Comparison of average distance travelled and speed while wearing different types of face mask compared with no mask

| Comparison | Average distance travelled (kilometres) with mask | Average distance travelled (kilometres), no mask | Estimated average difference in distance (km, 95% CI) | Average speed (km/hour) with mask | Average speed (km/hour), no mask | Estimated average difference in speed (km/hour, 95% CI) |
|---|---|---|---|---|---|---|
| *Distance travelled (km) during exercise (N=71)* | | | | | | |
| Cloth mask—no face mask | 2.75 | 2.82 | −0.07 (−0.15 to 0.01) | 11.06 | 11.37 | −0.31 (−0.64 to 0.02) |
| Surgical mask—no face mask | 2.78 | 2.82 | −0.04 (−0.12 to 0.04) | 11.27 | 11.37 | −0.10 (−0.43 to 0.22) |
| FFP3—no face mask | 2.76 | 2.82 | −0.06 (−0.14 to 0.02). | 11.12 | 11.37 | −0.24 (−0.57 to 0.08) |
| FFP, filtering face piece. | | | | | | |

cloth mask were ranked equal for ease of exercise and breathing through.

In free-text comments (online supplemental table 4), participants commented on the cloth mask becoming 'soaked through with sweat and water vapour'. A number commented on how the cloth mask 'sucks in when breathing in' or 'kept getting in my mouth', while others found it 'kept slipping down'. One participant described running in the cloth mask as 'like being waterboarded'. There were 35 negative free text comments about the cloth mask and one positive comment.

There were 15 negative free-text comments about the FFP3 mask, which some participants found 'constraining', 'so tight fitting it felt difficult to breathe' and caused 'nose pain'. Some people found this made them feel 'a bit dizzy', 'really lightheaded' or 'as though my heart was going really high when I was wearing it'. However, there were also 10 positive comments about wearing the FFP3, with these participants reporting 'a good fit', which 'was very comfortable', 'amenable to glasses-wearing' and which 'did not shake at all while running, unlike the others'.

There were 11 positive comments about wearing the surgical mask, mostly reporting it was 'fine', the 'easiest one' or caused 'little restriction'. The four negative comments about the surgical mask focused on glasses steaming up or the mask slipping.

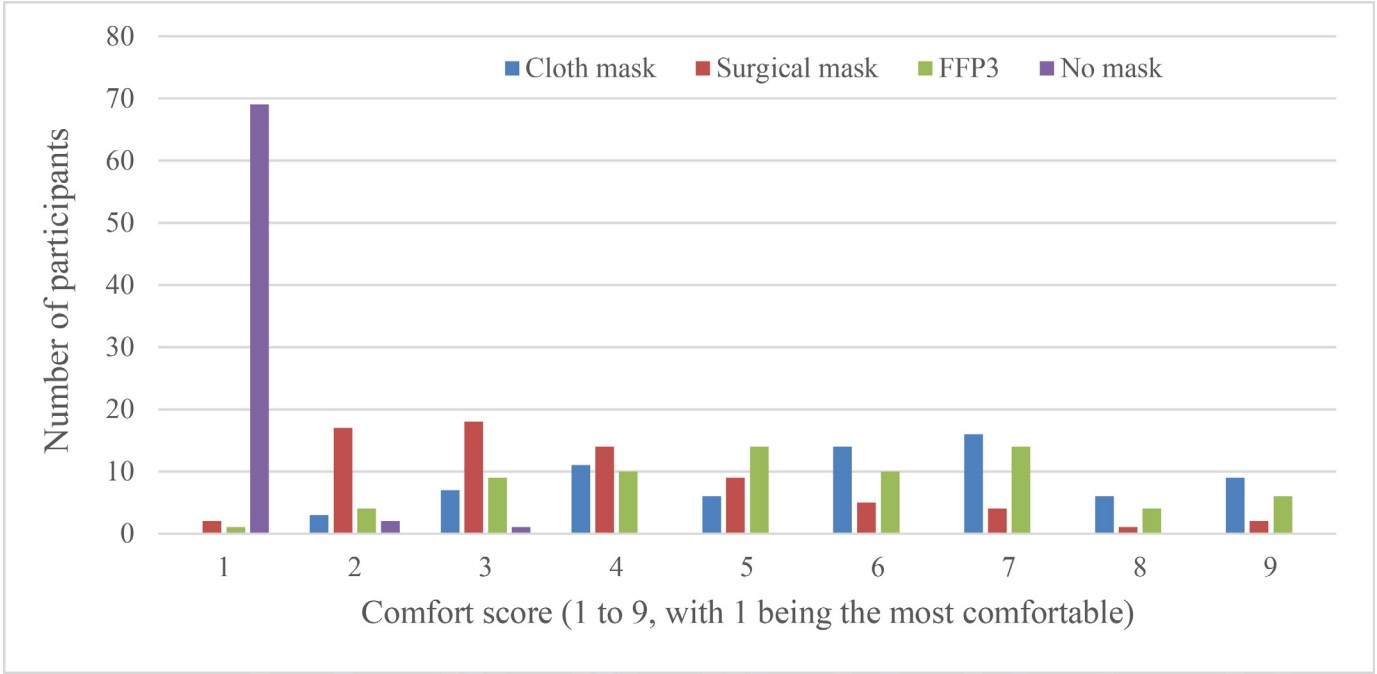

**Figure 1** Reported comfort of wearing different types of face mask during exercise. The graph plots the number of participants who reported each comfort score, between 1 and 9, for each of the three different types of face mask or no mask. FFP, filtering face piece.

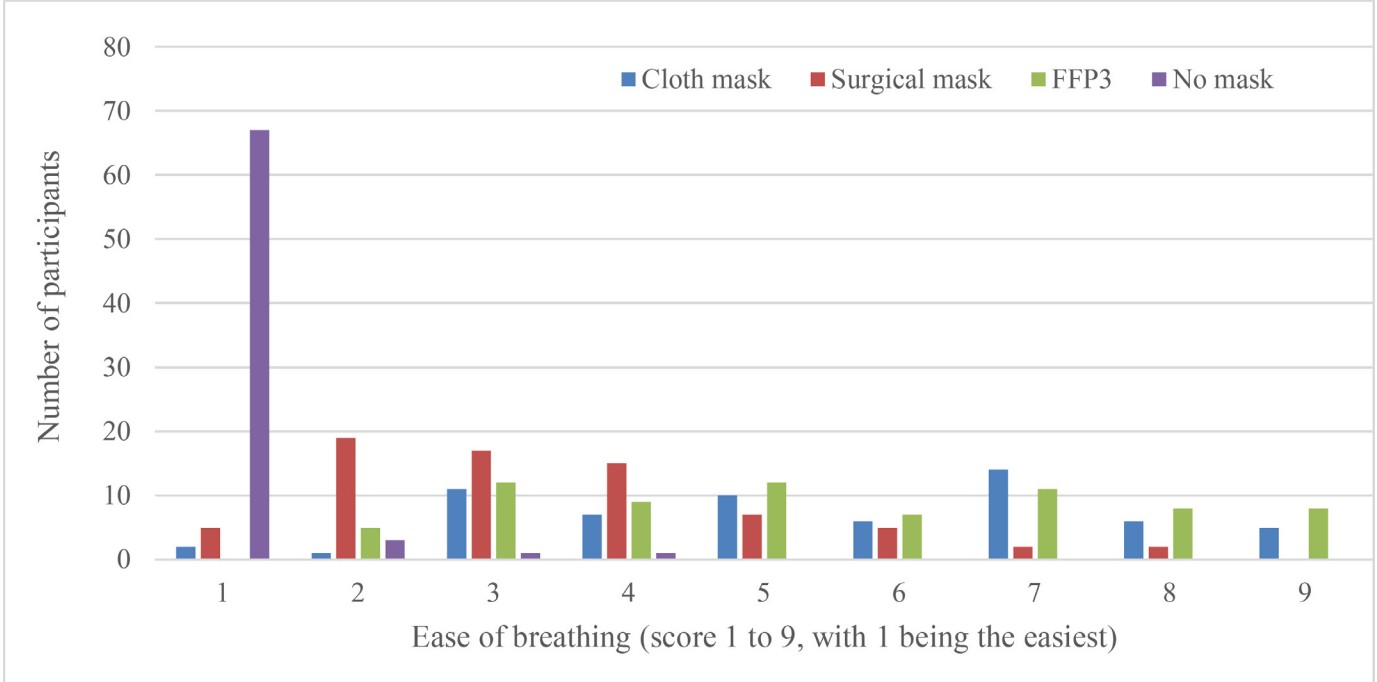

**Figure 2** Reported ease of breathing wearing different types of face mask during exercise. The graph plots the number of participants who reported each ease of breathing score, between 1 and 9, for each of the three different types of face mask or no mask. FFP, filtering face piece.

### Willingness to wear a face mask for exercise in the future

There were 33 participants who were accepting of the idea of mandatory face masks, 22 who were opposed and 17 who gave more neutral responses (online supplemental table 5). Positive replies included 'It would make no real impact to my exercise having to wear a face mask', 'Wouldn't bother me too much' and 'To be honest, if wearing a face mask during exercise will limit the spread of COVID-19 then I am happy to do so'. A number of participants mentioned they 'wouldn't mind doing so indoors', or 'only if it was the surgical mask'. In the neutral group there were comments such as 'I would tolerate it but not advocate it' and 'I would if I had to'.

Among those opposed people mentioned 'I would be annoyed, because I don't think that it's really needed for preventing transmission, especially outdoors' and that they 'would rather have a choice in the matter'. Some people compared favourably the experience of exercising with no mask compared with wearing any mask, with one saying 'When I ran with no mask at the end it felt like breathing in HD' (meaning high definition).

### DISCUSSION

We found no clinically significant reduction in $SaO_2$ or heart rate with any of the three face masks, suggesting

**Table 4** Comparison of reported comfort, ease of breathing and ease of overall exercise between the different types of face mask worn during exercise

| Type 1 | Type 2 | Number rating type 1 lower | Number rating type 2 lower | Probability type 1 rated lower (95% CI) | P value |
|---|---|---|---|---|---|
| *Ease of exercise, lower score=easier* | | | | | |
| Surgical* | FFP3 | 51 | 12 | 0.80 (0.73 to 0.86) | 0.00 |
| Surgical | Cloth | 50 | 12 | 0.81 (0.72 to 0.88) | 0.00 |
| Cloth | FFP3† | 29 | 31 | 0.49 (0.38 to 0.60) | 0.84 |
| *Ease of breathing, lower score=easier* | | | | | |
| Surgical* | FFP3 | 55 | 11 | 0.83 (0.76 to 0.88) | 0.00 |
| Surgical | Cloth | 59 | 10 | 0.86 (0.78 to 0.91) | 0.00 |
| Cloth | FFP3† | 28 | 36 | 0.44 (0.34 to 0.55) | 0.30 |

*Surgical is ranked first.
†FFP3 and cloth are equal second.
FFP, filtering face piece.

that exercising at moderate-to-high intensity wearing a face mask appears to be safe in healthy, young adults. However, around one in three participants reported additional symptoms when wearing a face mask, including feeling more breathless, dizzy or fatigued. Surgical face masks were experienced as the most comfortable, easiest to breathe through and had the least impact on exercise, though importantly we did not measure whether these masks were well-fitting and poor fit of surgical masks is common and greatly reduces protection.[16] There was most support for wearing a surgical face mask during indoor exercise if needed to reduce the spread of COVID-19, though around a third of participants were opposed to mandatory wearing of face masks during exercise.

## Strengths and limitations

To our knowledge, this is the largest study to date to investigate the safety and tolerability of wearing a face mask during exercise. We used a randomised cross-over design to ensure that the effect of any fatigue from exercise was equal between mask types. We asked participants to exercise at a consistent speed between study sessions and recorded their average speed and distance travelled, with no significant difference in exercise performance based on type of face mask worn.

We chose to recruit only healthy young participants to reduce the risk of serious adverse events, because of the limited pre-existing data demonstrating wearing a face mask during exercise was safe. However, our results may not be generalisable to people who have underlying health conditions or restricted exercise capacity and further research is needed in these populations.

We collected a limited amount of non-invasive physiological data from participants. Healthy young participants may maintain a constant $SaO_2$ and heart rate through compensatory mechanisms, such as increased respiratory rate, that we have not captured in our analysis. However, all of our participants were able to complete the exercise wearing each mask type with no significant change in their speed or distance travelled, supporting the finding that healthy young people can safely exercise wearing face masks.

Because of the COVID-19 pandemic and restrictions in social interaction, we conducted almost all of the study sessions outdoors. Study sessions took place across a range of seasons. During cold weather some participants had cold peripheries while exercising, meaning the pulse oximeters were not always able to record $SaO_2$ and heart rate or were slow to do so. Delays in obtaining accurate oximeter readings postexercise may have led to some recovery in heart rate and $SaO_2$ that would result in an underestimate of any change in these parameters. For these reasons, we also asked participants to fit their own face masks and we did not perform fitting tests. While this may have led to some discrepancy in mask fit between participants, it does reflect how face masks would be worn in practice. We had originally proposed to use FFP2 masks in our study protocol but there were supply issues that impacted us ordering these and we changed to use FFP3 masks instead.

## Comparison with other studies

Our results support the findings of a recent systematic review and meta-analysis, which assessed the impact of wearing a face mask during exercise.[17] This reported no significant difference in exercise performance among studies of surgical face masks (standard mean difference (SMD) −0.05, 95 CI −0.16 to 0.07) or N95 masks (−0.16, 95% CI −0.54 to 0.22).[17] Patients did have increased ratings of perceived exertion (0.33, 95% CI 0.09 to 0.58) and dyspnoea (0.60, 95% CI 0.30 to 0.90) with all masks.[17] In contrast to this review, we include cloth masks in the comparison of face masks types. The exercise tested in these studies was heterogeneous, ranging from 1-hour treadmill walks at low intensity, to strength training with half-squats and progressive exercise training on a bicycle. Our results demonstrate the principal findings of this review are applicable to healthy young adults during moderate-to-high intensity exercise.

A second recent systematic review of face mask wearing during rest, work or exercise identified 14 cross-over trials with 25 independent intervention arms.[18] There were a total of 246 participants across these studies. High intensity exercise was associated with a small reduction in $SaO_2$ (SMD −0.40, 95% CI −0.70 to −0.09) and lower respiratory rates (−0.25, 95% CI −0.44 to −0.06), particularly when wearing an FFP2 or N95 face mask.[18] However, there was no association with mask wearing and change in exercise performance, heart rate or exhaled $CO_2$. This review suggests that undertaking exhausting exercise wearing FFP-type face masks may reduce $SaO_2$ but supports our finding that mask wearing appears to be safe and does not negatively affect overall performance.

## Implications for policy and research

Governments globally are having to decide which public health measures to continue in response to changing incidence of COVID-19. Interventions that can contribute towards reducing transmission, while allowing people to return to normal life, are particularly important. Face masks can reduce transmission of the SARS-CoV-2 virus and appear to be safe to wear among young, healthy adults during exercise. Public health messaging could support their use in higher risk settings, such as indoor settings with poor ventilation or outdoor exercise among large volumes of people. Surgical masks appear to be the least restrictive type of mask to wear and may help reduce transmission by preventing the forward propulsion of a large jet of exhaled aerosol particles. Cloth masks and the FFP3 were ranked similarly by participants in our study in terms of ease of exercise and breathing through. This suggests that policy makers could consider recommending either surgical masks or the more secure FFP3, as opposed to cloth masks, should face masks be required during exercise. However, the comparative effectiveness of different mask types in reducing the risk

of transmission of SARS-CoV-2 does remains uncertain and could be a focus for future research. Future research could also examine the safety and tolerability of wearing a face mask during exercise among people with underlying health conditions, as results may differ from our healthy population. For example, a previous study of 97 people with chronic obstructive pulmonary disease performing a 6 min walking test wearing an N95 mask reported that those with more significant disease were more likely to experience dyspnoea or breathing discomfort.[19]

Wearing a face mask during exercise does not negatively impact exercise performance. It is possible there may even be benefits for longer-term endurance exercise performance if face mask wearing did induce mild hypoxia. Hypoxic exercise training has previously been reported to improve cardiac haemodynamics and exercise capacity,[20 21] though these studies typically recruited only small numbers of participants and have not been consistently replicated.[22] Further research into the long-term impact on exercise performance of wearing a face mask during exercise could help better inform athletes of the relative risks and benefits of this intervention.

## CONCLUSION

In this randomised cross-over trial, we found that healthy young adults were able to exercise at moderate-to-high intensity wearing either a surgical, cloth or FFP face mask without significant restriction on exercise performance or physiological limitation. However, wearing a face mask during exercise made some participants feel more breathless, dizzy or fatigued and most people felt it was less comfortable and enjoyable than exercising without a mask. There was limited support for wearing a face mask to exercise across all settings, but these results could support the use of face mask wearing while exercising in a limited number of high-risk settings as a safe intervention to reduce the risk of SARS-CoV-2 transmission.

**Acknowledgements** Thank you to all the volunteers who gave up their time to take part in this study. Thanks to Robert Temple who also helped with data collection on one study visit.

**Contributors** NJ and TG conceived the study. NJ, TG and HPD wrote the protocol. NJ, SM, KN and JB conducted all study visits. JO and NJ carried out the analysis. NJ wrote the first draft of the manuscript and is the guarantor and corresponding author. FDRH served as a scientific advisor. All authors contributed to the interpretation of the results and revisions of the manuscript. The corresponding author attests that all listed authors meet authorship criteria and that no others meeting the criteria have been omitted.

**Funding** Buff provided 60 original tubular cotton neckwear free of charge, which were used as the cloth masks. Buff had no role in study design, data collection, data analysis, data interpretation, writing of the report, or any other aspect of the study. NJ is supported by a Wellcome Trust Doctoral Research Fellowship (203921/Z/16/Z). FDRH acknowledges his part-funding from the NIHR Applied Research Collaboration (ARC) Oxford Thames Valley, the NIHR Oxford Biomedical Research Centre (BRC), and the NIHR Oxford Medtech and In-Vitro Diagnostics Co-operative. TG's research is supported by Oxford NIHR Biomedical Research Centre (grant no BRC-1215-20008).

**Competing interests** None declared.

**Patient and public involvement** Patients and/or the public were not involved in the design, or conduct, or reporting, or dissemination plans of this research.

**Patient consent for publication** Not applicable.

**Ethics approval** This study involves human participants and was approved by the University of Oxford Central University Research Ethics Committee (reference R&2015/RE001). Participants gave informed consent to participate in the study before taking part.

**Provenance and peer review** Not commissioned; externally peer reviewed.

**Data availability statement** Data are available on reasonable request.

**ORCID iDs**
Nicholas Jones http://orcid.org/0000-0002-0352-3785
Trisha Greenhalgh http://orcid.org/0000-0003-2369-8088

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
