## [Reviewer comments · BMJ Open]

ARTICLE DETAILS

TITLE (PROVISIONAL)	FaceMasks whilst Exercise Trial (MERIT): a crossover randomised controlled study
AUTHORS	Jones, Nicholas; Oke, Jason; Marsh, Seren; Nikbin, Kurosh; Bowley, Jonathan; Dijkstra, H Paul; Hobbs, Richard; Greenhalgh, Trisha

VERSION 1 – REVIEW

REVIEWER	Spang, Robert P. Technical University of Berlin, Quality and Usability Lab
REVIEW RETURNED	13-May-2022

GENERAL COMMENTS	First of all, thank you for your work! Here is a detailed list of concerns and suggestions. I hope they are providing helpful ideas to you. The manuscript leaves out critical information (descriptions regarding the instruments and how they were administered, e.g., heart rate or the questions asked). These are necessary to fully understand their procedure and to evaluate the soundness of the study: - Why did you recruit 72 participants when you registered only 60, and your power analysis (+ buffer) turned out only 60?- Pre-defined values (2% & 7bpm): where did you pre-define it? Is there e.g., any locked repository like the osf.io where you recorded this estimation before any data analysis? The study registration does not mention your pre-defined values.- You pre-registered FFP2 masks but used FFP3 without explaining why you changed your mind. What is the reason?- How was the during exercise measurement done? Did the participants stop their exercise for the measure? How was it conducted?- How and when was blood oxygenation measured? Which device?- p8 l25: "grouped into categories using a pre-defined framework" any information about that framework are necessary for evaluation- Did you adjust your alpha level for multiple comparisons in the post-hoc analyses? Referring to the difference between cloth masks and FFP3- Did you perform any fitting tests for the FFP masks?- A discussion of why to favor a non-inferiority schema over e.g., a TOST procedure would be interesting. Why did you choose non-inferiority, and why not "full" statistical equivalence?- p13 l51: The argument "... single session, minimizing the risk of recall bias when completing the study questionnaire." seems a bit far-fetched: In your within-subjects design, participants answered the questionnaire multiple times within only a few minutes. I'd
---

	argue that this especially allows for a recall bias. Hence, I suggest removing this argument.  - p13 l53: "We asked participants to exercise at a consistent speed" this is problematic since you investigated differences in traveled distance. Suppose participants were instructed to always run at the same speed. In that case, it should be expected to show similar distances throughout the conditions. This invalidates distance as an outcome variable. - p14 l9: as "further research" you could mention e.g., Kyung, S. Y., Kim, Y., Hwang, H., Park, J. W. & Jeong, S. H. Risks of N95 face mask use in subjects with COPD. Respir. Care 65 (5), 658–664 (2020) (doi.org/10.4187/respcare.06713). - The figures should rather be histograms or bar plots. Since there is no linear dependency between the variables, a line plot is not ideal for the data. # Problematic References  - The study with reference no. 7 (Fikenzer et al. 2022) is problematic, you might want to swap this one with another study. There are two letters to the editors arguing against its scientific soundness: Hopkins, S.R., Stickland, M.K., Schoene, R.B., et al. Effects of surgical and FFP2/N95 face masks on cardiopulmonary exercise capacity: the numbers do not add up. Clin Res Cardiol 109, 1605–1606 (2020). https://doi.org/10.1007/s00392-020-01748-0 and Kampert, M., Singh, T., Finet, J.E. et al. Impact of wearing a facial covering on aerobic exercise capacity in the COVID-19 era: is it more than a feeling?. Clin Res Cardiol 109, 1595–1596 (2020). https://doi.org/10.1007/s00392-020-01725-7 # Further suggestions  - p5 l16: Unclear where footnote 3 belongs to - p11 l30: I assume "other" symptoms" are described in the following two sentences, but I suggest pointing out clearly what they were - The finding "The FFP3 and cloth mask were ranked equal for ease of exercise and breathing through." could be discussed further: e.g., if participants rank them comparably, a policy recommendation could opt for the more secure FFP masks than for cloth masks. - On a side note: As you discuss further implications of wearing masks in everyday life, our work "The tiny effects of respiratory masks on physiological, subjective, and behavioral measures under mental load in a randomized controlled trial" might be interesting to you.
--	--

REVIEWER	Vipond, Joe University of Calgary, Emergency Medicine
REVIEW RETURNED	23-Jun-2022

GENERAL COMMENTS	ABSTRACT: As results section mention secondary outcomes (HR and tolerability), would suggest adding 2ary outcomes below the section stating primary outcome (O2 Sat) ABSTRACT: "...exercise the estimated average difference in oxygen saturations for cloth mask was -0.07% (95%CI -0.39 to 0.25), for surgical 0.28% (-0.04 to 0.60) and for FFP3 -0.21% (-0.53 to 0.11)."
--

	In body of paper these numbers are -0.07, -0.03, 0.00. Why the discrepancy? METHODS: In PARTICIPANTS a difference of 1.5% is stated as expected margin of difference, but in STATISTICAL ANALYSIS it states pre-specified non-inferiority of 2%. Why the difference? RESULTS: "...exercise wearing either a cloth mask (estimated average difference (EAD) - 0.78%, 95%CI -1.12 to -0.44) or the FFP3 compared to no mask (- 0.82%, -1.16 to -0.47), butnot when wearing the surgical mask (- 0.28%, -0.63 to 0.07)." These numbers don't match with the values presented in Table 1. They match up with this subgroup analysis though, which is confusing: " In a sub-group analysis among those with lower baseline SO₂, changes in SO₂ still did not exceed the non-inferiority margin of 2% although SO₂ during exercise was 1% lower among people wearing either the FFP3 (EAD -0.88, 95%CI -1.47 to -0.29) or cloth mask (- 0.90, -1.48 to -0.32)" Your table 1 and Supplemental table 1a are identical, but with different Ns and different titles. Could there be a confusion here? I suspect a major error in data transfer. I think table 1 needs to be reassessed for accuracy Table 1: If the number of participants is 72, why is the N 80, 78 and 74 in table 1? Perhaps a brief explanation Table 2: why is N=69 when participants are 72? Perhaps a brief explanation For tables: *FFP = filtering face piece respirator masks Instead of using asterisk to define FFP in tables, just put the definition in Box 1. TABLE 4: Consider switching position of FFP/Surgical (type 1/type 2) to make it easier to compare with Cloth/Surgical for both upper and lower sections. (0.80 vs. 0.81, and 0.83 vs. 0.86, instead of 0.20 vs. 0.81, and 0.17 vs. 0.86) 'When I ran with no mask at the end it felt like breathing in HD'. What is HD? "It is possible there may even be benefits for longer-term endurance exercise performance if facemask wearing did induce mild hypoxia." It would seem your study proves that this is not the case (there was no mild hypoxia induced by facemasks). Therefore this comment seems erroneously applied.
--	--

	Speed: although speed is mentioned as an outcome in the results, there is no data presented to affirm this result Consider changing abbreviation for Oxygen Saturation to O2Sat or SaO2 from SO2 (looks like sulfur dioxide, especially as you are using CO2 to mean Carbon Dioxide) https://www.allacronyms.com/oxygen_saturation/abbreviated
--	---

REVIEWER	Coleman, Brenda Sinai Health System, Infectious Disease Epidemiology Research Unit, University of Toronto
REVIEW RETURNED	12-Aug-2022

GENERAL COMMENTS	Limiting this to outdoor exercise during cold weather very much reduces the usefulness of the data. However, you did mention it in the discussion. Most jurisdictions did not require people to wear masks out of doors while most did require them in indoor settings. Page 5 (based on pagination in top left corner of proof): were participants required to keep the mask in place for the 1 minute after exercise was complete? What did participants do at the 15 minute mark? Sit down where they were (outside) or walk or ?? How did you measure distance? Please expand on the design/methods. Page 7: participants. I would exclude the rower (who completed their task indoors) out of the analysis. Should have 3 outcomes: ½ way vs base; completion vs base; 1 min post vs base Table 1: Comment on drop out during study....80 during/78 end/74 1 min after ***And, how did you have this many observations with only 72 participants? Table 1 information is the same as Table S1a – except for the number of participants. Also, the number of participants from Table S1a plus S1b does not sum to Table 1. You will need to review the text as well. Table 2: This may be an N issue, but it seems unlikely that you would have upper CIs over your non-inferiority margins for Tables S2a and S2b, but not for the full population for the FFP vs none comparison. sum of participants does not equal Table S2a plus S2b. The data is not duplicated, however as in Table 1. Page 10, lines59-60: you do not mention the data from Table S2b. Please do so. MINOR Move page 7, lines 19-31 to page 8. Perhaps line 7. It does not fit in participants section. You use “which” when the word “that” is correct. Page 1: Line 6, 40 and page 12 line 38 & 47; page 14 line 16, 31 & 38; other instances may occur, but I quit looking.
---

VERSION 1 – AUTHOR RESPONSE

Reviewer: 1

Dr. Robert P. Spang, Technical University of Berlin

Comments to the Author:

First of all, thank you for your work! Here is a detailed list of concerns and suggestions. I hope they are providing helpful ideas to you.

Thank you for taking the time to review our manuscript and for your very helpful comments.

The manuscript leaves out critical information (descriptions regarding the instruments and how they were administered, e.g., heart rate or the questions asked). These are necessary to fully understand their procedure and to evaluate the soundness of the study:

To address this point we have added an additional section in the Methods section, with the sub-heading 'Data collection', which reads as follows:

“SaO₂ and heart rate were measured using a finger pulse oximeter immediately prior to first commencing exercise, halfway through, on completion and 1 minute after completion for each bout of exercise. Participants paused their exercise at the midway point of exercise to collect these data. Participants continued to wear their facemask for one minute after completing the exercise, when the final recording for that bout of exercise was collected. Participants were asked to exercise at a consistent intensity between bouts of exercise. To confirm this was achieved, we measured the average speed and distance travelled during each bout of exercise using a Smartphone app (Strava) and assessed these data for significant differences across facemask type.

During each five-minute rest period the participants stood and completed a study questionnaire. This asked them to rate the comfort, ease of exercise and ease of breathing during exercise with the facemask they had just worn on a ten point Likert scale. At the end of the study session we also asked participants to identify the facemask that they found most difficult to exercise wearing, record any symptoms that they felt during exercise that were different to usual and free text comments regarding their willingness to exercise wearing a mask in the future (see Appendix 1).”

- Why did you recruit 72 participants when you registered only 60, and your power analysis (+ buffer) turned out only 60?

When conducting the study sessions we found that the finger pulse oximeters we were using to record oxygen saturations and heart rate would occasionally fail to provide a reading. This was not something that we had anticipated and we were concerned that the missing data would result in the study being under-powered to detect our primary outcome so continued to recruit beyond our additional intended sample size. In the manuscript tables we report the number of observations that were recorded. We have also added a comment on this problem with data collection in the limitations section of the manuscript:

“In a small number of participants at each time point it was not possible to obtain either a SaO₂ or heart rate reading on the pulse oximeter and the numbers reported in tables 1 and 2 reflect this.”

- Pre-defined values (2% & 7bpm): where did you pre-define it? Is there e.g., any locked repository like the osf.io where you recorded this estimation before any data analysis? The study registration does not mention your pre-defined values.

Although we registered the study on ClinicalTrials.gov, we did not publish our protocol in a locked repository or in the public domain. However, the non-inferiority margin for the primary outcome is included in the dated protocol that has been uploaded with this submission. This was the protocol that was reviewed by the University of Oxford Central University Ethics Research Committee.

- You pre-registered FFP2 masks but used FFP3 without explaining why you changed your mind. What is the reason?

At the time of ordering equipment we found that there were some limitations on the type of facemask available to us through our department, due to supply issues presumably caused by the COVID-19 pandemic. FFP3 masks were available and for the purpose of this study felt to be the best substitute mask type. We have added the following to the Limitations section:

“We had originally proposed to use FFP2 masks in our study protocol but there were supply issues that impacted us ordering these and we changed to use FFP3 masks instead.”

- How was the during exercise measurement done? Did the participants stop their exercise for the measure? How was it conducted?

Yes, participants stopped briefly to allow a member of the research team to record their heart rate and oxygen saturations using a finger pulse oximeter. We paused the timing of their running for this measurement. We have added additional information as follows:

“SaO₂ and heart rate were measured using a finger pulse oximeter immediately prior to first commencing exercise, halfway through, on completion and 1 minute after completion for each bout of exercise. Participants paused their exercise at the midway point of exercise to collect these data. Participants continued to wear their facemask for one minute after completing the exercise, when the final recording for that bout of exercise was collected.”

- How and when was blood oxygenation measured? Which device?

Please see the previous answer, where we outline the approach taken.

- p8 l25: "grouped into categories using a pre-defined framework" any information about that framework are necessary for evaluation

As mentioned in the Methods section, the initial framework categorised comments as either “positive, negative or neutral comments for each type of mask”. We have added a table of results showing how free text comments were categorised using this framework in Supplemental tables 4 and 5.

- Did you adjust your alpha level for multiple comparisons in the post-hoc analyses? Referring to the difference between cloth masks and FFP3

The confidence intervals presented in the main analysis do not adjust for multiple comparisons as this was not specified in our original analysis plan. We have now conducted a version of the main analysis which uses Dunnett’s method to adjust the confidence interval. This did not change the conclusions of our analysis and all comparisons remain non-inferior at the pre-specified margin.

- Did you perform any fitting tests for the FFP masks?

We did not perform any fitting tests for the FFP masks. We thought that fit testing would be unlikely to be done consistently if the general public were asked to wear facemasks for exercise so fit testing in the study would not reflect actual practice. We also aimed to minimise direct contact time between our research team and the participants because of the concern about the transmission risk of COVID-19 at the time of the study and therefore did not want to ask our study team to directly supervise any fit testing. We have added a comment on this in limitations section of the manuscript:

“For these reasons we also asked participants to fit their own facemasks and we did not perform fitting tests. Whilst this may have led to some discrepancy in mask fit between participants, it does reflect how facemasks would be worn in practice.”

- A discussion of why to favor a non-inferiority schema over e.g., a TOST procedure would be interesting. Why did you choose non-inferiority, and why not "full" statistical equivalence?

A non-inferiority hypothesis (i.e. a one-sided test) was preferred over an equivalence hypothesis (using two one sided tests or TOST) because we set out to establish whether wearing facemasks

during exercise was safe and were therefore primarily interested in detecting a significant drop in oxygen saturations or increase in heart rate. Furthermore, even before running the study and based on prior research we felt that it was likely that we knew the direction of effect from wearing a facemask; oxygen saturation would either remain unchanged or decrease, rather than increase and heart rate would remain unchanged or increase, due to its relationship with oxygen saturation..

- p13 l51: The argument "... single session, minimizing the risk of recall bias when completing the study questionnaire." seems a bit far-fetched: In your within-subjects design, participants answered the questionnaire multiple times within only a few minutes. I'd argue that this especially allows for a recall bias. Hence, I suggest removing this argument.

Thank you for this suggestion. We have removed this comment from the manuscript.

- p13 l53: "We asked participants to exercise at a consistent speed" this is problematic since you investigated differences in travelled distance. Suppose participants were instructed to always run at the same speed. In that case, it should be expected to show similar distances throughout the conditions. This invalidates distance as an outcome variable.

We apologise that this is unclear. We investigated differences in travelled distance and average speed only to confirm that participants had been able to adhere to the instruction of exercising at consistent speed across the bouts of exercise. We have removed distance as an outcome variable in recognition of the fact that it was included as a measure of control only. We have added a more detailed explanation to this end in the 'Study design' section of the Methods:

"Participants were asked to exercise at a consistent intensity between bouts of exercise. To confirm this was achieved, we measured the average speed and distance travelled during each bout of exercise using a Smartphone app (Strava) and assessed these data for significant differences across facemask type."

We have also added a comment in the results section when discussing the results of the comparison of average distance travelled:

"Across participants there was no significant difference in average speed or distance travelled when comparing any of the facemasks to no mask, *confirming participants had been able to exercise at a fairly consistent speed between bouts of exercise* (Table 3)."

- p14 l9: as "further research" you could mention e.g.,
Kyung,S.Y.,Kim,Y.,Hwang,H.,Park,J.W.&Jeong,S.H. Risks of N95 face mask use in subjects with COPD. *Respir. Care* 65 (5), 658–664 (2020) (doi.org/10.4187/respcare.06713).

Thank you, we have now added this reference along with a linked discussion point to our 'Implications for policy and research' section as follows:

"Future research could also examine the safety and tolerability of wearing a facemask during exercise among people with underlying health conditions, as results may differ from our healthy population. For example, a previous study of 97 people with chronic obstructive pulmonary disease performing a 6-minute walking test wearing an N95 mask reported that those with more significant disease were more likely to experience dyspnoea or breathing discomfort.²⁶"

- The figures should rather be histograms or bar plots. Since there is no linear dependency between the variables, a line plot is not ideal for the data.

Thank you for this suggestion. We have amended the figures to bar plots.

Problematic References

- The study with reference no. 7 (Fikenzer et al. 2022) is problematic, you might want to swap this one with another study. There are two letters to the editors arguing against its scientific soundness:

Hopkins, S.R., Stickland, M.K., Schoene, R.B., et al. Effects of surgical and FFP2/N95 face masks on cardiopulmonary exercise capacity: the numbers do not add up. *Clin Res Cardiol* 109, 1605–1606

(2020). <https://doi.org/10.1007/s00392-020-01748-0> and Kampert, M., Singh, T., Finet, J.E. et al. Impact of wearing a facial covering on aerobic exercise capacity in the COVID-19 era: is it more than a feeling?. Clin Res Cardiol 109, 1595–1596 (2020). <https://doi.org/10.1007/s00392-020-01725-7>
Thank you for bringing this to our attention. We have now removed reference to the Fikenzler study in our manuscript.

Further suggestions

- p5 l16: Unclear where footnote 3 belongs to

We apologise if this is not clear. We have removed all footnotes from the tables as per the recommendation of Reviewer 2 in an attempt to make this clearer.

- p11 l30: I assume " 'other' symptoms" are described in the following two sentences, but I suggest pointing out clearly what they were

Participants were asked to record any additional symptoms that they experienced during the study. We anticipated these might include symptoms such as claustrophobia or breathlessness so included these as tick boxes (see Appendix 1). However, we also recognised people may experience additional symptoms wearing a mask that we did not anticipate and so included a box to tick for these 'other' symptoms, with a comment box to record what these were. We have added a description of each of the 'other' symptoms that were recorded (see below). In addition we have included the study questionnaire as an Appendix and refer to this in the manuscript so that readers are able to access this if needed.

"The 'other' symptoms recorded as free-text comments were feeling light-headed, facial discomfort, skin irritation, a runny nose, pressure in the eyes, cramping and a stitch."

- The finding "The FFP3 and cloth mask were ranked equal for ease of exercise and breathing through." could be discussed further: e.g., if participants rank them comparably, a policy recommendation could opt for the more secure FFP masks than for cloth masks.

Thank you for this recommendation. We have added the following comment to the section on Implications for policy and practice:

"Cloth masks and the FFP3 were ranked similarly by participants in our study in terms of ease of exercise and breathing through. This suggests that policy makers could consider recommending either surgical masks or the more secure FFP3, as opposed to cloth masks, should facemasks be required during exercise."

- On a side note: As you discuss further implications of wearing masks in everyday life, our work "The tiny effects of respiratory masks on physiological, subjective, and behavioral measures under mental load in a randomized controlled trial" might be interesting to you.

Thank you for the recommendation and congratulations on the work. We read the manuscript with interest, particularly given the similarities with our work. We do feel trying to gather this objective evidence around the effects of facemask wearing is extremely important.

Reviewer: 2

Dr. Joe Vipond, University of Calgary

Comments to the Author:

ABSTRACT: As results section mention secondary outcomes (HR and tolerability), would suggest adding 2ary outcomes below the section stating primary outcome (O2 Sat)

Thank you we have added the secondary outcomes to the abstract (please see response to the Editor).

ABSTRACT: "...exercise the estimated average difference in oxygen saturations for cloth mask was -

0.07% (95%CI -0.39 to 0.25), for surgical 0.28% (-0.04 to 0.60) and for FFP3 -0.21% (-0.53 to 0.11)."

In body of paper these numbers are -0.07, -0.03, 0.00. Why the discrepancy?

Thank you, please see the comment below regarding Table 1. The figures in the abstract are correct and are now correctly reflected in those in Table 1.

METHODS:

In PARTICIPANTS a difference of 1.5% is stated as expected margin of difference, but in STATISTICAL ANALYSIS it states pre-specified non-inferiority of 2%. Why the difference?

We used an expected margin of difference of 1.5% for our power calculation based on the mean difference in oxygen saturations seen in the prior study cited in our manuscript. However, we felt that the non-inferiority margin for our study should be a difference that was both statistically and clinically significant and that this would be best achieved by setting this margin at a level that was a whole number that could be measured on a pulse oximeter to provide a more meaningful measure of difference. Of note, the results would still not have exceeded a non-inferiority threshold set at 1.5% for oxygen saturations.

RESULTS:

"...exercise wearing either a cloth mask (estimated average difference (EAD) - 0.78%, 95%CI -1.12 to -0.44) or the FFP3 compared to no mask (-0.82%, -1.16 to -0.47), but not when wearing the surgical mask (-0.28%, -0.63 to 0.07)."

These numbers don't match with the values presented in Table 1.

They match up with this subgroup analysis though, which is confusing:

" In a sub-group analysis among those with lower baseline SO₂, changes in SO₂ still did not exceed the non-inferiority margin of 2% although SO₂ during exercise was 1% lower among people wearing either the FFP3 (EAD -0.88, 95%CI -1.47 to -0.29) or cloth mask (-0.90, -1.48 to -0.32)"

Your table 1 and Supplemental table 1a are identical, but with different Ns and different titles. Could there be a confusion here? I suspect a major error in data transfer. I think table 1 needs to be reassessed for accuracy.

Table 1: If the number of participants is 72, why is the N 80, 78 and 74 in table 1? Perhaps a brief explanation.

We are extremely grateful to you for identifying the errors in Table 1. As you say, this reflects an error in the data transfer to table 1 in the manuscript. We have now corrected this and cross checked each of the values reported in the manuscript to ensure accuracy. We do apologise for this mistake.

Table 2: why is N=69 when participants are 72? Perhaps a brief explanation

In a small number of participants the pulse oximeters failed to provide a reading for either the oxygen saturations or pulse. The numbers reported in the table reflect the total number of results obtained at each time point. Table 1 has been corrected as per the comment above. We have added an explanation to this effect as follows:

"In a small number of participants at each time point it was not possible to obtain either a SaO₂ or heart rate reading on the pulse oximeter and the numbers reported in tables 1, 2 and the supplemental tables reflect this."

For tables: *FFP = filtering face piece respirator masks

Instead of using asterisk to define FFP in tables, just put the definition in Box 1.

Thank you for this helpful suggestion. We have updated the tables and footnotes accordingly.

TABLE 4: Consider switching position of FFP/Surgical (type 1/type 2) to make it easier to compare with Cloth/Surgical for both upper and lower sections. (0.80 vs. 0.81, and 0.83 vs. 0.86, instead of 0.20 vs. 0.81, and 0.17 vs. 0.86).

We have changed Table 4 as per your suggestion.

'When I ran with no mask at the end it felt like breathing in HD'.

What is HD?

High definition, meaning with heightened intensity. We have added an explanation of the abbreviation in the text:

"...*HD*' (meaning high definition)."

"It is possible there may even be benefits for longer-term endurance exercise performance if facemask wearing did induce mild hypoxia."

It would seem your study proves that this is not the case (there was no mild hypoxia induced by facemasks). Therefore this comment seems erroneously applied.

Whilst changes in oxygen saturation did not exceed our non-inferiority threshold, there was a small but statistically significant decrease in oxygen saturations during exercise wearing the FFP and cloth masks when compared to no mask. We accept that this comment is speculative but feel it is justified to state given we are highlighting the need for this to be further researched rather than stating it as a fact based on our results.

Speed: although speed is mentioned as an outcome in the results, there is no data presented to affirm this result

Thank you for this comment and please see the point above from Reviewer 1. Speed and distance were included as measures of control to check that participants had been able to comply with the instruction to try to exercise at a consistent intensity across the different bouts of exercise. We have added average speed to Table 3 to demonstrate there was no statistically significant difference in average speed for exercise wearing any facemask type compared to no mask.

Consider changing abbreviation for Oxygen Saturation to O2Sat or SaO2 from SO2 (looks like sulfur dioxide, especially as you are using CO2 to mean Carbon Dioxide)

https://www.allacronyms.com/oxygen_saturation/abbreviated

We have updated this abbreviation to SaO2 throughout the manuscript.

Reviewer: 3

Dr. Brenda Coleman, Sinai Health System, Infectious Disease Epidemiology Research Unit, University of Toronto

Comments to the Author:

Limiting this to outdoor exercise during cold weather very much reduces the usefulness of the data. However, you did mention it in the discussion. Most jurisdictions did not require people to wear masks out of doors while most did require them in indoor settings.

We agree that this is a limitation of the study and had originally intended to run more sessions indoors but were prevented from doing so by the restrictions on indoor social interaction in place in the UK at the time. However, we would point out that the study was conducted between June and January and most of the participants exercised in the autumn when the weather was relatively mild. We have made clearer in the limitations section that study sessions were held across a range of seasons:

“Study sessions took place across a range of seasons.”

Page 5 (based on pagination in top left corner of proof): were participants required to keep the mask in place for the 1 minute after exercise was complete? What did participants do at the 15 minute mark? Sit down where they were (outside) or walk or ?? How did you measure distance? Please expand on the design/methods.

Thank you for these comments. This was also recommended by Reviewer 1 and we have added an additional section in the Methods to explain in more details how the study was conducted (please see pervious comment for the full text added).

Page 7: participants. I would exclude the rower (who completed their task indoors) out of the analysis. Should have 3 outcomes: ½ way vs base; completion vs base; 1 min post vs base

Thank you for this suggestion. We have included a sensitivity analysis excluding the rower from the analysis of changes in oxygen saturations and heart rate and this does not significantly alter the results (Supplemental tables 3a and b).

Our primary analysis is a comparison of different types of facemask compared to no mask and we felt that this was best assessed using a non-inferiority analysis comparing the difference in estimated average oxygen saturations between each type of mask to no mask at each time point, rather than a comparison within each mask type against baseline oxygen saturations.

Table 1: Comment on drop out during study....80 during/78 end/74 1 min after

***And, how did you have this many observations with only 72 participants?

Table 1 information is the same as Table S1a – except for the number of participants.

Also, the number of participants from Table S1a plus S1b does not sum to Table 1.

You will need to review the text as well.

Thank you for highlighting the error in Table 1. Please see the previous response to Reviewer 2. We apologise again for not detecting this error ourselves before submitting the manuscript.

Table 2: This may be an N issue, but it seems unlikely that you would have upper CIs over your non-inferiority margins for Tables S2a and S2b, but not for the full population for the FFP vs none comparison.

We agree that this is likely to reflect the relatively small number of participants in the sub-group analysis, which has resulted in wide confidence intervals. We have highlighted the small sample size in the results section now:

“However, the upper 95% CI did exceed the non-inferiority margin of 7bpm for the FFP3 mask compared to no mask at each time point measured, *although the sample size for this analysis was small.*”

The sum of participants does not equal Table S2a plus S2b. The data is not duplicated, however as in Table 1.

Please see previous comments to Reviewer 2. There were some missing data as a result of a small number of instances where the finger pulse oximeter did not detect any oxygen saturation or heart rate reading. We have added the following comment:

“In a small number of participants at each time point it was not possible to obtain either a SaO₂ or heart rate reading on the pulse oximeter and the numbers reported in tables 1, 2 and the supplemental tables reflect this.”

Page 10, lines59-60: you do not mention the data from Table S2b. Please do so.

We have added a description of the results from Table S2b, with the updated text in italics as follows:

“Among the subgroups with SaO₂ ≤ 98% or SaO₂ > 98% *at baseline*, there was no significant difference in average heart rate during or after exercise with any facemask (Supplemental tables 2a and 2b).

However, for the subgroup with SaO₂ ≤ 98% the upper 95% CI did exceed the non-inferiority margin of

7bpm for the FFP3 mask compared to no mask at each time point measured, *although the sample size for this analysis was small*. This was also true for the surgical mask at the end of exercise and after a 1 minute rest period (*Supplemental table 2a*). *For the subgroup with SaO₂ > 98% the upper 95% CI exceeded the non-inferiority margin for the FFP3 and surgical masks compared to no mask during exercise only (Supplemental table 2b).*"

MINOR

Move page 7, lines 19-31 to page 8. Perhaps line 7. It does not fit in participants section.

Thank you for this suggestion. We have moved the section as suggested.

You use "which" when the word "that" is correct. Page 1: Line 6, 40 and page 12 line 38 & 47; page 14 line 16, 31 & 38; other instances may occur, but I quit looking.

We have read through the manuscript again and changed from 'which' to 'that' in a number of places.

We (respectfully!) disagree that all uses of the word 'which' are incorrect.

VERSION 2 – REVIEW

REVIEWER	Spang, Robert P. Technical University of Berlin, Quality and Usability Lab
REVIEW RETURNED	18-Oct-2022

GENERAL COMMENTS	Dear team, thank you for considering most of my previous remarks and also plenty of the other reviewers'. The paper in its current form is good and should be published. Thank you for your work.
---

REVIEWER	Vipond, Joe University of Calgary, Emergency Medicine
REVIEW RETURNED	04-Oct-2022

GENERAL COMMENTS	overall excellent! Thanks for fixing the issues pointed out in the first review. Only one minor comment: For lines 251-252 and 285-288: For example, a previous study of 97 people with chronic 286 obstructive pulmonary disease performing a 6-minute walking test wearing an N95 mask 287 reported that those with more significant disease were more likely to experience dyspnoea or 288 breathing discomfort" "This reported no significant 251 difference in exercise performance among studies of surgical facemasks (standard mean 252 difference (SMD) -0.05, 95CI -0.16 to 0.07) or N95 masks" Consider changing to "FFP-equivalent" or "respirator-style mask" from "N95" as these are the only references to N95 in your manuscript and you have not defined the term previously in the paper.
---

REVIEWER	Spang, Robert P. Technical University of Berlin, Quality and Usability Lab
REVIEW RETURNED	18-Oct-2022

GENERAL COMMENTS

Dear team, thank you for considering most of my previous remarks and also plenty of the other reviewers'. The paper in its current form is good and should be published. Thank you for your work.